# Treatment of Pediatric Acute Lymphoblastic Leukemia: A Historical Perspective

**DOI:** 10.3390/cancers16040723

**Published:** 2024-02-08

**Authors:** Hiroshi Hayashi, Atsushi Makimoto, Yuki Yuza

**Affiliations:** 1Department of Hematology/Oncology, Tokyo Metropolitan Children’s Medical Center, 2-8-29 Musashidai, Fuchu 183-8561, Tokyo, Japan; atsushi_makimoto@tmhp.jp (A.M.); yuki_yuza@tmhp.jp (Y.Y.); 2Department of Laboratory Medicine, Tokyo Metropolitan Children’s Medical Center, 2-8-29 Musashidai, Fuchu 183-8561, Tokyo, Japan

**Keywords:** acute lymphoblastic leukemia, chemotherapy, clinical trials

## Abstract

**Simple Summary:**

The treatment of pediatric acute lymphoblastic leukemia (ALL) has improved dramatically over the past half-century thanks to the refinement of risk stratification and risk-adapted therapies. Conventional chemotherapeutic agents with optimized treatment intensity have cured a substantial proportion of children with ALL. The present review discusses the history of developmental therapeutics for pediatric ALL in various countries through an extensive literature review and proposes a model for a treatment backbone for pediatric ALL, which may serve both as a guide for the treatment of ALL in low- and middle-income countries and as a foundation for future, international clinical trials.

**Abstract:**

Acute lymphoblastic leukemia (ALL) is the most common disease in pediatric oncology. The history of developmental therapeutics for ALL began in the 1960s with the repetition of “unreliable” medical interventions against this lethal disease. By the 1990s, the development of multi-agent chemotherapy and various types of supportive care rendered ALL treatable. Highly sophisticated, molecular, diagnostic techniques have enabled highly accurate prediction of the relapse risk, and the application of risk-adapted treatments has increased the survival rate in the standard-risk group to nearly 100% in most European nations and North America. Incorporation of state-of-the-art, molecularly targeted agents and novel treatments, including cell and immunotherapy, is further improving outcomes even in the high-risk group. On the other hand, the financial burden of treating children with ALL has increased, imperiling the availability of these diagnostic and treatment strategies to patients in low- and middle-income countries (LMICs). The fundamental treatment strategy, consisting of corticosteroid and classical cytotoxic therapy, has achieved fairly good outcomes and should be feasible in LMICs as well. The present review will discuss the history of developmental therapeutics for childhood ALL in various countries through an extensive literature review with the aim of proposing a model for a treatment backbone for pediatric ALL. The discussion will hopefully benefit LMICs and be useful as a base for future clinical trials of novel treatments.

## 1. Introduction

Acute lymphoblastic leukemia (ALL) is the most common pediatric malignancy [1]. While ALL was incurable just a half-century ago, the discovery of new drugs and the optimization of treatment protocols have led to a dramatic improvement in the survival rate of children with ALL [2]. During the last decade, the five-year survival of children aged 0–14 years exceeded 90% in most high-income countries [3]. Figure 1 shows the chronological improvement in overall survival in children with ALL in the United States. This improvement in survival was realized by the development of new treatments based on knowledge accumulated through careful observation of individual cases by numerous pediatric oncologists and the fruits of several clinical trials carried out by various cancer cooperative study groups across the world (Figure 2), such as the Children’s Oncology Group (COG)—a merger of the Children’s Cancer Group (CCG) and the Pediatric Oncology Group (POG)—and the Berlin–Frankfurt–Münster (BFM) study group, among others [4]. Intensification of treatment for the population at high risk of a relapse and the avoidance of treatment-related toxicity in the low-risk population were the guiding principles of these clinical trials.

On the other hand, patient survival remains substantially lower in low- and middle-income countries (LMICs), where multiple factors such as a high frequency of toxicity-related deaths and treatment discontinuation remain challenges to effective ALL treatment [5]. Moreover, costly new drugs, such as bi-specific T-cell engagers and chimeric antigen receptor (CAR) T cells, and cutting-edge diagnostic tests, including the evaluation of minimal residual disease (MRD) by polymerase chain reaction (PCR) or multicolor flow cytometry, which have contributed to the recent increase in the survival rate, have yet to become available in all countries. However, it should be noted that conventional, cytotoxic chemotherapies with appropriate supportive therapy have successfully cured a substantial proportion of children with ALL in the United States and Europe.

The present study aims to clarify the key components of ALL treatment, which may be feasible even in regions with limited access to high-end therapies, and to present a fundamental treatment structure for ALL treatment that may be used as a common platform upon which to perform future clinical trials globally.

## 2. The Dawn of Chemotherapy

In 1948, Farber et al. first reported temporary remissions of acute leukemia in five patients following the administration of aminopterin, a folic acid antagonist [6]. Farber discovered that the use of folic acid, which was indicated for pernicious anemia, unexpectedly resulted in leukemia progression. From this observation, he hypothesized that folic acid stimulated the growth of leukemia cells and experimented with ways to block this pathway [7]. Independently, Hitchings and Elion created 6-mercaptopurine (6-MP) to interfere with DNA metabolism [7]. The clinical effectiveness of a combination therapy consisting of two antimetabolites, 6-MP and methotrexate (MTX), against leukemia was demonstrated by Frei et al. in 1961 [8]. Subsequently, several novel, antileukemic drugs, including vincristine (VCR), asparaginase (ASP), cyclophosphamide (CPM), daunorubicin (DNR), and cytarabine (Ara-C), were introduced in the 1960s. Vincristine, an extract of the periwinkle plant, was first used as a monotherapy and produced transient, complete remission in approximately half of the patients tested [9,10]. The combination of these antileukemic drugs with adrenal corticosteroid further improved the rate of remission induction in patients with ALL.

The “Total Therapy” studies by Pinkel et al. effected a major breakthrough in the development of treatments for ALL [11]. “Total therapy” consisted of several, essential treatment components, such as remission induction, intensification therapy, central-nervous-system (CNS)-directed therapy, and continuous maintenance therapy, each requiring one to two months to complete and all of which still serve as essential components in the current treatment for ALL. In Total Therapy Study V, which began in 1967, more than 50% of the enrollees remained in remission after treatment completion and had favorable, long-term survival [12]. This breakthrough made ALL a curable disease for the first time [13].

## 3. History of Risk Stratification

Risk-adapted therapy, which has been incorporated into current strategies for treating ALL, augments therapy in patients with a poor prognosis while helping to avoid over-treatment in those with a good prognosis [14]. The so-called BFM risk factor in the BFM-81 study was a stratification system based on a retrospective analysis of the BFM 70/76 study, which incorporated the leukemia cell count in peripheral blood and hepatomegaly and splenomegaly at diagnosis to stratify risk [15]. These factors were considered to reflect the tumor burden; patients with a high tumor burden had a greater risk of relapse. In the United States, a uniform risk classification known as the National Cancer Institute (NCI) criteria was used to define the standard-risk and high-risk groups [16]. The standard-risk (SR) category included children aged 1 to 9 years and a white blood cell (WBC) count less than 50,000/microliter while including the other patients in the high-risk (HR) category (Table 1). Age and WBC count have been recognized as independent prognostic factors in most clinical trials [14]. Besides age and WBC count at diagnosis, CNS involvement and male sex were identified as clinically adverse prognostic factors [17].

Based on immunophenotype, ALL can be categorized as either B-cell ALL (B-ALL) or T-cell ALL (T-ALL). T-ALL accounts for approximately 15% of ALL cases. Historically, the outcomes of T-ALL have been inferior to those of B-ALL. However, treatment intensification by risk stratification has narrowed the gap between the groups. Since 1996, both B-ALL and T-ALL have been treated in accordance with the NCI criteria, as exemplified in the CCG trial. Treatment of SR T-ALL in the CCG-1952 and 1991 trials demonstrated results similar to those of POG-9404, which was performed specifically for T-ALL [18]. Today, the use of nelarabine and Capizzi methotrexate has further improved the survival of children with T-ALL [19].

Various cytogenetic and molecular abnormalities can serve as prognostic factors in patients with ALL. Cytogenetic technologies, such as karyotyping and fluorescence in situ hybridization (FISH), are used to evaluate gross chromosomal alterations. Secker-Walker et al. first demonstrated the prognostic value of changes in chromosome number in ALL, which suggested that high hyperdiploidy (HHD) was a marker of a favorable prognosis [20]. HHD is the most common cytogenetic abnormality observed in childhood B-cell precursor ALL and accounts for up to 35% of cases [21]. Various studies demonstrated an excellent outcome, and treatment reduction became possible for this subgroup [22,23]. Conversely, while hypodiploidy is uncommon, it is associated with a very poor prognosis [24]. Various structural chromosomal abnormalities, including translocations, deletions, insertions, and inversions, which are usually associated with specific genetic abnormalities, have been identified in most ALL cases as well [25]. For example, ALL with cryptic translocation of t(12;21)(p13;92), which generates the *ETV6*::*RUNX1* fusion gene, is one of the most common subtypes of childhood ALL and has been shown to have a favorable prognosis [17].

Finally, the in vivo response to treatment has become a significant prognostic factor. The ALL-BFM 83 study revealed that patients with fewer than 1000 blasts/mm^3^ in their peripheral blood after a prednisone pre-phase (“prednisone good responders” or PGR) had far better outcomes than their counterparts (“prednisone poor responders” or PPR), leading to the adoption of the prednisone response as a stratification factor in BFM-86 [26]. In the CCG-141 trial, M3 bone marrow on Day 14 was found to be associated with a lower complete remission (CR) rate [27]. In addition to morphological analysis of peripheral blood or bone marrow smears, MRD measured with PCR or flow cytometry has proved to be a significant prognostic factor optimizing post-induction therapy [28,29]. The Recife RELLA05 pilot study in Brazil demonstrated that a simplified MRD assessment with flow cytometry was efficacious in identifying very low-risk patients and feasible in LMICs [30].

Each study group has a specific risk classification system, all of which are being continuously amended in the light of new findings. Table 1 presents examples of risk classification systems for ALL that have been used in recent clinical trials.

## 4. The Backbone of ALL Treatment

### 4.1. Remission Induction Therapy

Prior to the establishment of clinical study groups, chemotherapeutic agents for remission induction therapy were chosen on the basis of clinical observation or small clinical studies, as described in the previous chapter. In the 1950s, the combination of MTX and 6-MP was often used for this purpose [31]. Data on the remission induction rate associated with the use of various agents, which had been accumulated by the middle of 1960s, demonstrated that VCR and prednisone were more effective than the other agents available at the time [32,33]. Later, Acute Leukemia Group B demonstrated that a combination of VCR and prednisone (VP) produced the highest complete remission rate (84%) [34]. In 1967, L-asparaginase (L-ASP) was introduced into clinical use and proved to be highly effective when combined with VP [35].

During this search for the optimal combination therapy, Riehm et al. had already begun to use what would become the backbone of contemporary remission induction therapy when they introduced an eight-week induction protocol (protocol I) with the eight most effective antileukemic drugs available at the time in the West-Berlin Therapy study [36,37]. The main concept underlying the treatment was an increase in the intensity of the treatment to the limits of individual tolerance [38]. Protocol I consisted of two, distinct phases; the first half (protocol Ia) included prednisone, ASP, VCR, and DNR [39]. Glucocorticoid, VCR, and ASP exert antileukemic effects without a strong myelosuppressive effect, thus serving as excellent agents for remission induction in various treatment protocols [40]. The BFM trials experimented with decreasing the anthracycline dosage. The ALL-BFM 90 trial found that reducing the cumulative anthracycline dosage from 160 mg/m^2^ to 120 mg/m^2^ of body surface area did not adversely affect the prognosis [41]. The ALL-BFM 95 trial enrolling patients with SR achieved favorable results while reducing the number of daunorubicin doses during induction therapy to two doses of 30 mg/m^2^.

In the US, the CCG-105 trial was the first to incorporate a BFM-based regimen [42]. For children younger than 10 years, a three-drug induction regimen without daunorubicin in conjunction with protocol II provided an event-free survival (EFS) rate comparable to that achievable by a four-drug induction regimen. The three-drug induction resulted in fewer days of hospitalization and became the standard for later versions of the CCG and COG regimens. In the CCG-106 trial for HR ALL, a BFM-76-based regimen demonstrated superiority over the New York regimen and became the basis of the treatment protocol for HR ALL in the US [43].

Historically, both BFM and CCG/COG studies have shown that glucocorticoid, VCR, and ASP were the three key drugs in induction therapy for SR ALL and that anthracycline can be omitted or reduced without changing the treatment efficacy in this subgroup (Figure 3). Malaysia–Singapore ALL 2003 demonstrated that MRD-based risk stratification enabled anthracyclines to be omitted from induction therapy without compromising EFS [44]. On the other hand, a four-drug induction regimen including anthracycline remains the gold standard for HR ALL treatment.

### 4.2. Consolidation Therapy

The second half of Protocol I (Protocol Ib), also called consolidation therapy, consisted of CPM, Ara-C, 6-MP, and intrathecal MTX, all of which had been adopted in the chemotherapeutic panoply since the publication of the BFM 70/76 trial results [36,39]. The CCG-105 trial compared a BFM-based consolidation therapy (intensive consolidation) to the standard, less-toxic consolidation therapy comprising only 6-MP, vincristine, and intrathecal (IT) MTX [42]. The results demonstrated that these two regimens produced similar EFS rates. Later CCG trials for SR ALL included only VCR, 6-MP, and IT MTX in the consolidation phase [45]. In the CCG-1882 and 1961 trials, augmented consolidation consisting of the standard BFM consolidation therapy (Protocol Ib) with CPM, Ara-C, and 6-MP together with an additional phase including VCR and pegylated (PEG) ASP improved EFS in patients with HR ALL [46,47] but failed to do so in patients with SR ALL in the AALL0331 trial [23]. Augmented consolidation again failed to show any advantage over the standard consolidation regimen in either intermediate-risk (IR) or HR ALL patients in the ALL IC-BFM 2009 trial [48]. Moreover, a new combination of CPM and etoposide for HR ALL failed to demonstrate a higher survival rate than that seen in consolidation therapy with CPM/Ara-C/6-MP in the AALL1131 trial [49]. Overall, 6-MP-based consolidation therapy is undoubtedly essential to ALL therapy, and the standard, BFM consolidation therapy comprising CPM, Ara-C, and 6-MP is probably too toxic, at least for patients with SR ALL. Although the standard BFM consolidation regimen is commonly used for HR ALL, the value of augmented consolidation is still controversial.

### 4.3. Interim Maintenance Therapy

In terms of interim maintenance therapy (IM), the CCG-1882 trial introduced the so-called Capizzi MTX, which consists of increasing the IV MTX dosage from 100 to 300 mg/m^2^ without leucovorin rescue [46]. The result was improved five-year EFS in HR patients with a slow, early response. The CCG-1991 trial for SR ALL demonstrated the superiority of intensified IM using a Capizzi-like regimen (a combination of VCR and escalating dose IV MTX without ASP) to the conventional IM using 6-MP and PO MTX [45]. In the AALL0232 trial, HD-MTX was superior to Capizzi MTX for HR B-ALL [50] whereas in the AALL0434 trial, Capizzi MTX was more effective for T-ALL [51]. BFM-90 and BFM-95 intensified protocol M in an MR group by combining HD-MTX with L-ASP and Ara-C, respectively, but failed to improve the outcomes [41,52]. In the 2009 ALL IC-BFM trial, MTX 2 g/m^2^ was as effective as 5 g/m^2^ for IR and HR-ALL [48]. The advantage of using MTX 2 g/m^2^ is that it can be safely administered even in institutions where the blood serum concentration of MTX cannot be promptly measured [53]. Although there is as of yet no direct comparison of HD-MTX and Capizzi MTX for SR ALL, the current evidence supports the use of HD-MTX for HR B-ALL and Capizzi MTX for T-ALL.

### 4.4. Reinduction Therapy

The BFM 76/79 trial first introduced reinduction therapy (Protocol II) following remission induction therapy (Protocol Ia and Ib), which successfully improved survival in patients with HR ALL [54]. Protocol II was basically a repetition of Protocol I with some changes to avoid drug resistance, such as replacing prednisone with dexamethasone or replacing daunorubicin with doxorubicin. In the BFM 79/81 trial, reduced-intensity, postinduction intensification (Protocol III) was also introduced into the treatment of SR ALL [55]. However, the addition of Protocol III soon after remission induction therapy showed no advantage. In the same study, delayed administration of Protocol II for HR ALL was more effective and less toxic than immediately administering Protocol II after Protocol I. After the BFM-81 trial, reinduction therapies were designed to be performed in all risk groups after interim maintenance therapy consisting of MTX and 6-MP. In order to lessen treatment-induced toxicity, BFM-83 attempted to omit Protocol III from SR ALL treatment, but the outcomes were significantly poor and led to reaffirming the clinical significance of reinduction therapy for SR ALL [37]. Two major study groups, the Associazione Italiana di Ematologia e Oncologia Pediatrica (AIEOP) and BFM, jointly conducted the AIEOP-BFM ALL 2000 trial, which randomly allocated SR ALL patients to Protocol II or Protocol III and demonstrated that the reduction of chemotherapy in Protocol III resulted in an increased rate of relapse [56]. For IR ALL, Protocols II and III—each administered twice—were shown to have an equivalent treatment effect [57].

The CCG-105 trial for IR ALL demonstrated a survival advantage of delayed-intensification (DI) therapy similar to that achieved by Protocol II [42]. The CCG-1891 trial for IR ALL demonstrated that doubling the DI resulted in better survival than single DI [58]. However, the CCG-1961 trial found no benefit of doubling the DI for HR-ALL [47]. Moreover, the CCG-1991 study reported similarly that doubling the DI conferred no benefit on patients with SR ALL [45]. Hence, it should be noted that doubling the DI is not always advisable for either SR or HR ALL in the framework of the current ALL treatment strategy.

Many clinical trials de-intensifying reinduction therapy for SR ALL have produced conflicting results. In contrast to the AIEOP-BFM ALL 2000 trial, the Ma-Spore-ALL 2010 trial using a historical cohort demonstrated that the complete omission of anthracycline was noninferior to a protocol that included anthracycline during delayed intensification [59]. Future studies need to examine the issue of optimizing the intensity of reinduction therapy, especially for SR ALL.

### 4.5. Maintenance Therapy

Despite the fact that VCR and prednisone induced complete remission in more than 80% of the patients tested, merely continuing this treatment was insufficient to maintain the initial remission [9,60]. A clinical trial conducted by the Southwestern Oncology Group (SWOG) using one of three oral maintenance therapies, 6-MP, MTX, or CPM [61], was the first trial to demonstrate the efficacy of maintenance therapy.

The optimal duration of maintenance therapy depends on the backbone and intensity of the preceding chemotherapy. The BFM 81/83 trial, which randomized patients to a short maintenance therapy or a long maintenance therapy group (total treatment duration of 18 months and 24 months, respectively), demonstrated the superiority of long maintenance therapy [62]. However, the AIEOP 1979 trial series, which compared a treatment duration of two and three years, found no advantage in extending maintenance therapy [63]. A subset of pediatric patients with ALL (female sex; *TCF3*::*PBX1*, *ETV6*::*RUNX1*) in the Tokyo Children’s Cancer Study Group (TCCSG) L92-13 trial achieved a cure after six months of maintenance therapy, although other patients in the trial did not [64].

Historically, in some cooperative study groups, including COG, male patients have undergone treatment longer than female patients to mitigate the normally inferior outcomes among male patients [65]. However, with intensification and prior refinement of treatment, prolonging maintenance therapy has become less effective in narrowing the survival gap between the sexes, and most contemporary clinical trials implemented an equal maintenance therapy duration.

Monthly VCR plus corticosteroid pulse therapy, which is characteristic of the current COG regimen, first proved its efficacy in the CCG-161 trial [66]. However, the benefit of pulse therapy was not evident in the context of an intensified BFM regimen for IR ALL in the I-BFM-SG ALL IR 95 trial [67]. The ALL-B12 trial, a nationwide, Japanese study incorporating a BFM backbone, also demonstrated that VCR/dexamethasone intensification failed to improve survival in patients with SR ALL [68]. In the AALL0932 trial, patients receiving VCR/dexamethasone pulses every four or every twelve weeks had the equivalent EFS and overall survival (OS) [69]. A meta-analysis demonstrated that less frequent pulse therapy did not negatively affect survival in recent trials while more frequent pulse therapy was associated with increased toxicity [70]. Taken together, monthly VCR and corticosteroid pulse therapy may be omitted from maintenance phase without worsening the outcomes.

## 5. Other Elements in ALL Treatment

### 5.1. Choice of Glucocorticoids

Dexamethasone has a longer plasma and cerebrospinal fluid half-life and better CNS penetration than prednisone [71,72]. The Cancer and Leukemia Group B (CALGB, formerly known as Acute Leukemia Group B) first compared prednisone 40 mg/m^2^/day and dexamethasone 6 mg/m^2^/day during remission–induction and maintenance therapy and demonstrated significantly fewer CNS relapses in the dexamethasone arm [73]. The CCG-1922 trial, a larger randomized control trial conducted by CCG [74], produced similar results. The AIEOP-BFM 2000 trial also demonstrated the superiority of dexamethasone 10 mg/m^2^/day to prednisone 60 mg/m^2^/day in a three-week course of induction therapy in terms of EFS, although induction-related mortality was higher in the dexamethasone arm [75]. Subgroup analysis of the study revealed the greatest benefit in patients with T-ALL and those responsive to prednisone, although the rate of salvageable relapses was likely lower. At a prednisone-to-dexamethasone dosage ratio less than 7, dexamethasone achieved better EFS than prednisone, although adverse effects including infection, bone fracture, and mood and behavior problems occurred more frequently [72]. The Dana-Farber Cancer Institute (DFCI) ALL Consortium Protocol 91-01 and St. Jude Total Therapy Study XIIIB study also suggested that administering dexamethasone only in post-remission therapy may be a way of improving EFS and CNS control [76,77].

In terms of safety, the incidence of serious bacterial and fungal infections was significantly higher among patients receiving dexamethasone in the AIEOP-BFM 2000 trial [75]. The risk was especially high for patients older than 10 years and led to a stoppage of the enrollment of patients in this age group. The AALL0232 trial demonstrated a higher incidence of osteonecrosis in children older than 10 years receiving dexamethasone 10 mg/m^2^/day on days 1 to 14 than in their counterparts receiving prednisone 60 mg/m^2^/day on days 1 to 28 during induction [50]. The CCG-1961 trial demonstrated that alternate-week dexamethasone administration during delayed intensification was effective in decreasing the osteonecrosis incidence associated with dexamethasone [78]. Generally, children older than 10 years have a higher risk of dexamethasone-associated adverse events, which can be attributed to a slower rate of dexamethasone clearance than seen in younger patients [79]. Although dexamethasone is apparently more effective than prednisone, dosage adjustments are necessary to avoid treatment-related toxicity, especially in teenagers.

### 5.2. Asparaginase Therapy

In 1967, Hill et al. reported the first case series to demonstrate the therapeutic effect of *Escherichia coli* (*E. coli*) L-ASP and even reported complete bone marrow remission in one patient [80]. Three formulations of ASP are currently available, including native *E. coli* L-ASP, pegylated *E. coli* asparaginase (pegaspargase), and *Erwina chrysanthemi* asparaginase (Erwinase). The dosing and administration schedule for ASP varies widely among study groups. In a randomized controlled trial conducted by CCG, pegaspargase demonstrated more rapid clearance of lymphoblasts than native *E. coli* L-ASP [81]. Although pegaspargase is more expensive than native L-ASP per vial and may be unavailable in some regions, pegaspargase may be considered more cost-effective because it is associated with a lower rate of hypersensitivity reactions [82].

Discontinuation of ASP therapy has been repeatedly shown as a factor negatively affecting survival [76,83]. The optimal duration of ASP therapy is controversial; however, based on the result of The Nordic Society of Paediatric Haematology and Oncology (NOPHO) ALL2008 trial, it can be assumed that 10 to 16 weeks are optimal for non-HR ALL patients [84,85]. Further, the Dutch Childhood Oncology Group (DCOG) ALL-11 demonstrated that a median pegaspargase dosage of 450 IU/m^2^ was able to maintain a trough level >100 IU/L, which is considered sufficient to deplete L-asparagine [86].

### 5.3. Intrathecal Therapy

Intrathecal therapy is one of the key modalities for preventing CNS recurrences of ALL. In Total Therapy Study V, five doses of IT MTX successfully decreased the number of patients with a CNS relapse to three of the 35 patients enrolled in the study [12]. The CCG-1952 trial for SR ALL demonstrated that intrathecal triple therapy (ITT) decreased CNS relapses to a greater degree than IT MTX, whereas bone marrow relapses paradoxically increased and OS was inferior [87]. Further, the AALL1131 trial demonstrated that ITT was not superior to single-agent IT MTX for HR ALL [88]. Given these results, the standard of care for intrathecal therapy remains IT MTX.

Regarding the initial IT therapy schedule, a POG study demonstrated that ITT starting on day 1 of systemic treatment increased mortality mainly due to infections during induction [89]. The Taiwan Pediatric Oncology Group (TPOG) ALL-2002 trial also demonstrated that delaying initial IT therapy after the clearance of blasts in peripheral blood or performing it within ten days of the start of induction therapy did not worsen the outcomes and eliminated traumatic lumber punctures with blasts [90]. The initial IT therapy does not need to be administered on the first day of systemic therapy but rather when the patient is clinically stable within a week or so from the start of induction therapy.

The total number of IT therapies largely varies among the studies. The recently published IC-BFM 2009 trial included 17 to 21 IT treatments and CRT for patients with CNS-3 [48]. The COG trial involved the prolongation of IT therapy during maintenance therapy. For example, the AALL0932 trial for SR ALL included 11 IT therapies before maintenance therapy, with IT therapy continuing to be administered every 12 weeks during maintenance therapy [69]. The St. Jude Total Therapy Study XVI, which completely omitted CRT, included 13 to 21 IT treatments for low-risk ALL and 16 to 27 treatments for SR ALL [91]. In the context of CNS-directed therapy without CRT, frequent and intensive IT therapy extending to the period of maintenance therapy is required regardless of the type of backbone therapy.

### 5.4. Cranial Radiotherapy

In the early days of the development of ALL treatment, CNS relapse was the main reason for disrupting complete remission initially induced by multi-agent chemotherapy. Total Therapy Study V first achieved a long-term survival rate exceeding 50% through introducing 24 Gy cranial radiotherapy (CRT) and IT MTX administration [12]. Prophylactic CRT was necessary to prevent CNS leukemia when systemic therapy was inadequate to control the disease [92]. However, the CRT led to neurocognitive impairment, endocrinopathy, and secondary CNS malignancies [93,94,95,96], and modern ALL protocols tend to omit prophylactic CRT. The introduction of high-dose methotrexate (HD-MTX) and intensive IT therapy allowed cranial radiation to be removed entirely from BFM 86/90 for SR ALL, and from BFM 95 for SR ALL and medium risk (MR) ALL (except T-ALL) [37,52]. CRT was limited to patients with a high risk of CNS relapse or an overt CNS disease at the initial diagnosis; a T-cell immunophenotype; a high initial WBC count; or a slow, early response [62]. Total Therapy Study XV demonstrated that CRT can be safely omitted without increasing the risk of a CNS relapse in children with newly diagnosed ALL by intensifying systemic treatment with MTX, dexamethasone, ASP, and IT therapy [97]. In a meta-analysis combining data from ten cooperative study groups, CRT was found to have minimum impact on reducing the risk of a relapse in children treated with a contemporary protocol [98]. Although CRT has contributed to improving the treatment outcomes of CNS leukemia, its complete removal is becoming a de facto standard in current ALL treatments.

## 6. Management of ALL in Low- and Middle-Income Countries

Although 80% of newly diagnosed, pediatric cancer cases worldwide occur in LMICs, their outcomes are suboptimal for various reasons compared to those in high-income countries (HICs) [99]. Factors such as the inability to access cancer care facilities; delays in initiating treatment after symptom onset; discontinuation of treatment, such as maintenance therapy, especially in the outpatient setting; and inadequate supportive care have been identified as responsible for decreased overall survival in patients with pediatric ALL [100,101]. Many LMICs have limited resources for laboratory and radiological diagnostic equipment, which hinders or delays the accurate diagnosis of pediatric malignancies including ALL [102]. In particular, multiparametric flow cytometry for immunophenotyping and MRD assessment may not be available in LMICs, thus posing a major barrier to the diagnosis and risk stratification of ALL [103]. From the therapeutic perspective, unlike for other pediatric solid tumors, neither surgery nor radiotherapy is generally required for the treatment of ALL, although complex chemotherapy still requires experienced medical staff and relevant supportive therapies. Infection is a major cause of mortality during ALL treatment in LMICs; a report from a Vietnamese hospital found that 43% of deaths were caused by infection [104].

Despite numerous social and economic problems surrounding medical care systems, the outcome of pediatric ALL is gradually improving both in LMICs and HICs. Establishing a dedicated pediatric oncology unit significantly improved five-year EFS in patients with pediatric ALL from 32% to 63% in the resource-poor city of Recife in Brazil [105]. Multidisciplinary teams consisting of specially trained staff, compliance to protocol-based therapy, and locally funded family support were described as key factors leading to their success. From the perspective of the management of treatment-related toxicity, reducing treatments in a population with a low risk of relapse while enhancing supportive care is especially required in LMICs [106,107]. Preventing infections involves avoiding excess myelosuppression and establishing timely access to diagnostics and antimicrobials [108]. Collaboration with major academic centers in developed countries may help find a solution to delays in diagnosis and ways to improve the evaluation of the treatment response, including MRD measurements [109]. All these measures have contributed and will continue to contribute to further improving the survival of children with ALL in LMICs.

## 7. Future Directions

The treatment of pediatric ALL has advanced dramatically over the past half century and may be considered one of the most successful episodes in the battle against cancer in both adults and children. The present review article attempted to outline the progress in treatment strategy while discussing improvements in the survival rate in terms of the choice and dosing of the key drugs in treatments for ALL. The treatment protocol of the latest clinical trial conducted in high-income countries is neither necessarily the best treatment option nor feasible due to issues of drug availability and safety. Reflecting on the development of ALL treatments is important for elucidating the core elements of ALL treatments, allowing LMICs to achieve dramatic improvements in the clinical management of pediatric ALL.

Different cooperative study groups have adopted varying treatment strategies based on their individual treatment philosophy. The risk stratification system, and the combination of cytotoxic agents and their dosages, vary widely among the groups. Nevertheless, these groups share a common principle rooted in the early days of the development of chemotherapy, and multiple trials in different study groups have shown some findings to be reproducible.

In addition to the differences in treatment protocol, variations in patient characteristics must be taken into account. Children with ALL vary in numerous ways, including age, sex, race, drug metabolism, physical strength, psychosocial condition, etc., and these variations impact the efficacy of treatments. Thus, this review aimed to present the fundamental structure of ALL treatments abstracted from different clinical trials, which may be applied with maximum safety and efficacy regardless of the patients’ profile or availability of medical resources. Figure 3 demonstrates the fundamental treatment structure based on the research findings described above. Of course, refining the sequence and combination of cytotoxic agents may further improve the survival of children with ALL.

Reducing treatment-related, long-term comorbidities is another issue that needs to be solved. Although fewer patients today have comorbidities than in the era before the introduction of risk-stratified therapy, there is still room for improvement, particularly in high-risk patients [110]. The incidence of secondary neoplasms, especially the more aggressive forms, remains significantly higher than in the general population [111]. While the risk of cardiomyopathy has decreased thanks to restrictions on the use of anthracyclines, major joint replacement and diabetes are emerging as adverse effects of increased corticosteroid and ASP use [110,112]. Continuous, retrospective monitoring of sequential treatment regimens is vital for further improving ALL treatment.

To tackle these unanswered clinical questions, international collaboration involving researchers from both high-income countries and LMICs is desirable. Narrowing the survival gap in children with ALL between high-income countries and LMICs should be the goal of future research. It is to be hoped that the present article will help pediatric oncologists worldwide in obtaining a deeper understanding of the essential aspects of the structure of treatments for pediatric ALL.

## 8. Conclusions

The treatment of pediatric ALL has evolved through numerous clinical trials conducted over the past decades and has dramatically improved patient survival. The combination of classical chemotherapy with optimized treatment intensity by risk level has cured most cases. The next step would be to reduce the treatment burden in SR ALL while augmenting the treatment of HR ALL with novel therapeutics. At the same time, the development of less toxic novel agents is desirable to minimize safety issues, including late and long-term effects. The knowledge gained from this historical process may serve to improve the survival of children with ALL in LMICs and help to design future clinical trials around the world.

## Figures and Tables

**Figure 1 cancers-16-00723-f001:**
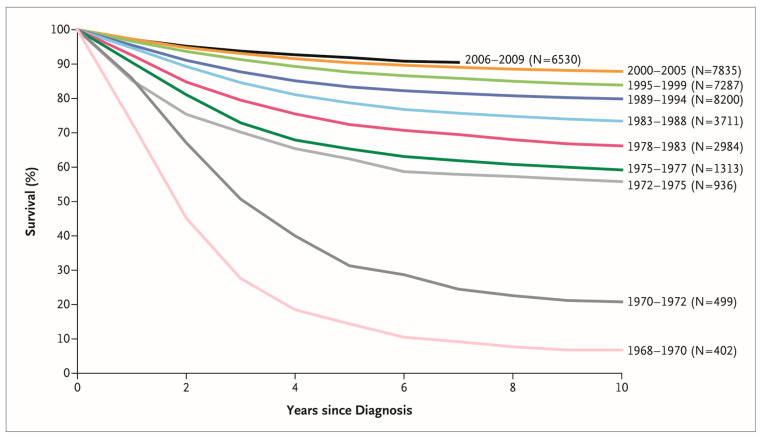
Chronological improvement in overall survival of children with acute lymphoblastic leukemia enrolled in Children’s Cancer Group and Children’s Oncology Group clinical trials. Reprinted with permission from “Acute Lymphoblastic Leukemia in Children,” by Stephen P. Hunger and Charles G. Mullighan, 2015, *N. Engl. J. Med.*, *373* (16), 1541–1552. Copyright (2015) by Massachusetts Medical Society.

**Figure 2 cancers-16-00723-f002:**
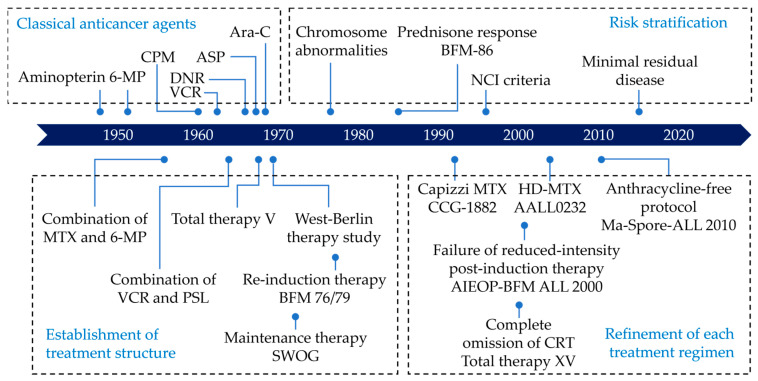
Timeline of the development of ALL treatment. This figure was designed by the authors. ALL: acute lymphoblastic leukemia; ASP: asparaginase; Ara-C: cytarabine; CPM: cyclophosphamide; DNR: daunorubicin; HD-MTX: high-dose methotrexate; MTX: methotrexate; PSL: prednisone; VCR: vincristine; 6-MP: 6-mercaptopurine.

**Figure 3 cancers-16-00723-f003:**
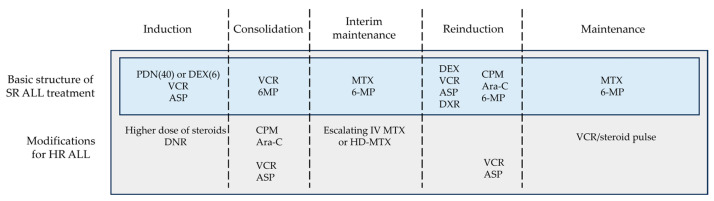
Basic structure of ALL treatment. This figure was designed by the authors. ALL: acute lymphoblastic leukemia; ASP: asparaginase; Ara-C: cytarabine; CPM: cyclophosphamide; DEX: dexamethasone; DNR: daunorubicin; DXR: doxorubicin; HD-MTX: high-dose methotrexate; HR: high-risk; IV: intravenous; MTX: methotrexate; PDN: prednisone; SR: standard-risk; VCR: vincristine; 6-MP: 6-mercaptopurine.

**Table 1 cancers-16-00723-t001:** Example of risk stratification system in recent BFM and COG clinical trials.

BFM	SR	IR		HR	
	1. PB day 8: blasts < 1000/μL 2. Age 1–6 years 3. Initial WBC < 20,000/μL 4. If available FC MRD < 0.1% or M1/M2 marrow on day 15 5. No M2/M3 marrow on day 33 *All criteria must be fulfilled.*	All patients not stratified as SR or HR are intermediate-risk patients		1. IR and if available FC MRD >10% or M3 marrow on day 15 2. SR if available FC MRD >10% 3. PB on day 8: blasts ≥ 1000/μL 4. M2/M3 marrow on day 33 5. Translocation t(9;22) [*BCR*::*ABL*] or t(4;11) [*KMT2A*::*AFF1*] 6. Hypodiploidy ≤ 44 *At least one criterion must be fulfilled.*	
COG	SR			HR	
	Low SR	Average SR	High SR	Age ≥ 10 years and/or WBC ≥ 50,000/µL	
	Age 1.0–9.99 years, WBC < 50,000/µL				
	Triple trisomy OR *ETV6::RUNX1* and Day 8 or 15 marrow M1 and Day 29 marrow M1 and Day 29 MRD < 0.1% and no CNS 2/3 or testicular disease	No triple trisomy OR *ETV6::RUNX1* and Day 8 or 15 marrow M1 and Day 29 marrow M1 and Day 29 MRD < 0.1%	ANY patient with CNS 3 or testicular disease OR Day 15 marrow M2/M3 OR Day 29 MRD ≥ 0.1–1% OR *KMT2A* translocation with a RER OR steroid pretreatment (select cases)	Day 8 or 15 marrow M1 and Day 29 marrow M1 and Day 29 MRD < 0.1% and no CNS 3 or testicular disease	Day 15 marrow M2/M3 OR Day 29 MRD ≥0.1–1%, OR CNS 3 or testicular disease OR *KMT2A* translocation with a RER OR steroid pretreatment (select cases)

The BFM stratification system was adopted from the IC-BFM 2009 trial. The COG stratification system was adopted from the AALL0331 and AALL0232 trials. M1, <5% blasts; M2, 5–25% blasts; and M3, >25% blasts. BFM: Berlin–Frankfurt–Münster study group; COG: Children’s Oncology Group; FC: flow cytometry; HR: high risk; IR: intermediate risk; MRD: minimal residual disease; PB: peripheral blood; RER: rapid early response; SR: standard risk; WBC: white blood cell.

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
