# Peer review of "Treatment of Pediatric Acute Lymphoblastic Leukemia: A Historical Perspective"

_cancers, 2024, doi:10.3390/cancers16040723_

Round 1

Reviewer 1 Report

Comments and Suggestions for Authors

The review is interesting and provide a useful overview on pediatric ALL treatment and its progress. However, some important points (and, then, sections) are missing in my opinion. Here are my main comments.

-       Introduction: Even though the authors clearly cite the source, are they authorize to reproduce this figure in this paper? I mean, they should confirm that there are no copyright issues with it.

-       A figure summarizing the main diagnostic-therapeutic achievements in the medical management of pediatric ALL would add interest to this article. Maybe also including the main protocols.

-       Some aspects related to radiotherapy should be explained, in order to describe its very limited use nowadays compared to the past. Indeed, this aspect would also be mentioned in a description of potential long-term sequelae of the treatment, which is not the main challenge since a very good survival has been obtained, at least in high-income countries (see: Best Pract Res Clin Haematol. 2022 Dec;35(4):101403. doi: 10.1016/j.beha.2022.101403; others.). A specific section should be created.

-       In the introduction, the authors mention potential issues in treatment of ALL outside high-income countries. A specific section should discuss these aspects (e.g. doi: 10.1097/CCO.0000000000000125). Anyway, some barriers are related not only to therapeutic aspects, but also to diagnostic possibilities, which are common across different pediatric malignancies, even if here these limitations could be more impactful, due to the absence of standardized protocols, unlike ALL (as discussed in: Front Oncol. 2022 Oct 6;12:985862. doi: 10.3389/fonc.2022.985862)

-       Figure 2 should be placed before in my opinion.

-       After the future directions, a conclusion section should be added.

-       References should be expanded and completed to support the new sections, according to the previous suggestions.

-       The numeration of the sections should be revised. For instance, in my opinion, the general description of ALL therapeutic management, should be one section and the different phases should be included as different subsections.

Comments on the Quality of English Language

See above

Reviewer 2 Report

Comments and Suggestions for Authors

This is a very interesting review of the treatment of paediatric acute lymphoblastic leukaemia, considering the improvements achieved over time. I have only a few suggestions to improve the manuscript as below reported:

Lines 47-49: in this sentence is not clear the meaning of “high-risk or low-risk population”, please explain

Line 146: explain what is meant by "CR rate". I suggest that the explanation of the abbreviation also be included in the text and not only in the footnotes of Table 1.

Lines 152-153: Table 1 should be mentioned before, when the data are described in the text. The table should also be better organised to make it easier to read.

Similarly to what was done for risk stratification (Table 1), I suggest that the authors also make similar tables for the other chapters, in particular for remission induction therapy, consolidation therapy, interim maintenance therapy, re-induction therapy and maintenance therapy. This should make it easier for the reader to follow the various steps reached in the different trials.

Round 2

Reviewer 1 Report

Comments and Suggestions for Authors

Reviewer's comments:

The review is interesting and provide a useful overview on pediatric ALL treatment and its progress. However, some important points (and, then, sections) are missing in my opinion. Here are my main comments.

RR- UNFORTUNATELY, THE REVISED MANUSCRIPT HAS NOT BEEN SUBMITTED IN WORD-REVIEW MODE (WHICH CAN BE ALSO PRINTED IN PDF) AND THE REVIEWER CANNOT EASILY CHECK THE CHANGES. MOREOVER, SEVERAL AND IMPORTANT POINTS HAVE NOT BEEN DEVELOPED BY THE AUTHORS (see -RR)

Introduction: Even though the authors clearly cite the source, are they authorize to reproduce this figure in this paper? I mean, they should confirm that there are no copyright issues with it.

Thank you for your comments. We were granted the right to reproduce the figure by the publisher. The license details are described in a separate file (“permission.v1.pdf”).

RR-ACCEPTED

A figure summarizing the main diagnostic-therapeutic achievements in the medical management of pediatric ALL would add interest to this article. Maybe also including the main protocols.

Figure 2 was intended to give a simple summary of the medical management of pediatric ALL. Because our protocol documents are written in Japanese and need a long time to translate, we chose not to submit at this time.

RR- UNFORTUNATELY, THE AUTHORS HAVE NOT ADDRESSED AT ALL THIS IMPORTANT POINT. AGAIN, I RECOMMEND TO WORK ON THIS POINT ACCORDING TO THE PREVIOUS RECOMMENDATION.

Some aspects related to radiotherapy should be explained, in order to describe its very limited use nowadays compared to the past. Indeed, this aspect would also be mentioned in a description of potential long-term sequelae of the treatment, which is not the main challenge since a very good survival has been obtained, at least in high-income countries (see: Best Pract Res Clin Haematol. 2022 Dec;35(4):101403. doi: 10.1016/j.beha.2022.101403; others.). A specific section should be created.

[Response]

We have created a specific section discussing the radiotherapy in “5.4 Cranial Radiotherapy”.

RR- ACCEPTED

In the introduction, the authors mention potential issues in treatment of ALL outside high-income countries. A specific section should discuss these aspects (e.g. doi: 10.1097/CCO.0000000000000125). Anyway, some barriers are related not only to therapeutic aspects, but also to diagnostic possibilities, which are common across different pediatric malignancies, even if here these limitations could be more impactful, due to the absence of standardized protocols, unlike ALL (as discussed in: Front Oncol. 2022 Oct 6;12:985862. doi: 10.3389/fonc.2022.985862)

[Response]

We have created a specific section discussing the ALL treatment in LMICs in “6 Management of ALL in low and middle-income countries”.

RR- AS PREVIOUSLY MENTIONED, THE AUTHORS SHOULD ALSO HIGHLIGHT THE DIFFERENCES WITH OTHER PEDIATRIC MALIGNANCIES. INDEED, AS CORRECTLY STATED BY THE AUTHORS, THE PROGNOSIS OF PEDIATRIC ALL IS GRADUALLY IMPROVING IN LMCIs, BUT SOME DIAGNOSTIC BARRIERS STILL EXIST AND ARE SHARED WITH OTHER MALIGNANCIES, ESPECIALLY THE RAREST ONES, ALSO IN TERMS OF INSTRUMENTAL STAGING AND IMMUNO-GENETIC ANALYSES. THEREFORE, I RECOMMEND TO DEVELOP MORE THESE POINTS, ACCORDING TO THE PREVIOUS SUGGESTIONS.

Figure 2 should be placed before in my opinion.

[Response]

Figure 2 summarizes the treatment of pediatric ALL. Thus, we consider it is more appropriate to place Figure 2 in the current section.

RR- I DISAGREE WITH THE AUTHORS’ RESPONSE. INDEED, THE GENERAL SCHEME OF PEDIATRIC ALL TREATMENT IS DESCRIBED IN SECTION 4 (NOT SECTION 7).

RR- AS AN ADDITIONAL COMMENT AND POINT, SINCE THIS MANUSCRIPT ALSO AIMS AT PROVIDING AN “HISTORICAL PERSPECTIVES”, I THINK THAT A TIMELINE OF THE MAIN PROTOCOLS IMPROVEMENTS COMING FROM THE SEVERAL NATIONAL/INTERNATIONAL SOCIETIES COULD BE INCLUDED IN THE DISCUSSION AND A NEW FIGURE MAY SUMMARIZE IT.

After the future directions, a conclusion section should be added.

[Response]

We have created a conclusion section.

RR- ACCEPTED IN GENERAL. HOWEVER, I SUGGEST DEVELOPING IT A LITTLE MORE, SINCE IT IS TOO GENERIC IN THE CURRENT VERSION.

References should be expanded and completed to support the new sections, according to the previous suggestions.

[Response]

We have expanded the references according to your advice.

RR- TO BE FURTHER IMPROVED ACCORDING TO THE PREVIOUS SUGGESTIONS AND ADDITIONAL DISCUSSION POINTS, TO BE DEVELOPED YET.

The numeration of the sections should be revised. For instance, in my opinion, the general description of ALL therapeutic management, should be one section and the different phases should be included as different subsections.

[Response]

We have placed the discussion of different phases of ALL treatment in subsections under “Backbone of ALL treatment”.

RR- ACCEPTED.

RR-I WOULD ASK THE AUTHORS TO PROVIDE THE NEXT REVISION IN WORD-REVIEW MODE. THE SIMPLE YELLOW-HIGHLIGHTING IS NOT APPROPRIATE.

Comments on the Quality of English Language

SOME IMPROVEMENTS ARE NEEDED AND, ALSO FOR THIS ASPECT, A WORD-REVIEW MODE REVISED MANUSCRIPT IS REQUESTED.

Round 3

Reviewer 1 Report

Comments and Suggestions for Authors

THE AUTHORS FURTHER IMPROVED THE MANUSCRIPT. I WOOULD SUGGEST SOME MINOR SPECIFICATIONS AND TOPIC COMPLETION, AS EXPLAINED BELOW.

In the introduction, the authors mention potential issues in treatment of ALL outside high-income countries. A specific section should discuss these aspects (e.g. doi: 10.1097/CCO.0000000000000125). Anyway, some barriers are related not only to therapeutic aspects, but also to diagnostic possibilities, which are common across different pediatric malignancies, even if here these limitations could be more impactful, due to the absence of standardized protocols, unlike ALL (as discussed in: Front Oncol. 2022 Oct 6;12:985862. doi: 10.3389/fonc.2022.985862)

[Response1]

We have created a specific section discussing ALL treatment in LMICs in “6 Management of ALL in low and middle-income countries”.

RR- as previously mentioned, the authors should also highlight the differences with other pediatric malignancies. indeed, as correctly stated by the authors, the prognosis of pediatric all is gradually improving in lmcis, but some diagnostic barriers still exist and are shared with other malignancies, especially the rarest ones, also in terms of instrumental staging and immuno-genetic analyses. therefore, i recommend to develop more these points, according to the previous suggestions.

[Response2]

We have added a discussion of the diagnostic barriers in LMICs that are common to ALL and other pediatric malignancies as well as those that are specific to ALL. (p.10, lines 426-433)

RRR- I THANK THE AUTHORS FOR BETTER DISCUSSING THIS IMPORTANT ASPECT, WHICH SHOULD BE INDEED HIGHLIGHTED IN SUCH A COMPREHENESIVE, BUT GENERAL, NARRATIVE REVIEW. SO, THE MAIN PROBLEMS/BARRIERS ARE NOW CLEAR ENOUGH, BUT I WOULD SUGGEST THIS ARTICLE PROVIDES SOME CURRENT AND/OR POTENTIAL SOLUTIONS, AS DISCUSSED IN ONE OF THE ARTICLES MENTIONED IN THE PREVIOUS COMMENTS, WHERE THE AUTHORS EXPERIENCED A COOPERATION WITH MAJOR CANCER ACADEMIC CENTERS IN DEVELOPED COUNTRIES FOR ADDITIONAL ANALYSES OR INTERPRETATIONS, FOR INSTANCE, EVEN IF THIS IS DONE IN THE FIELD OF A DIFFERENT MALIGNANCY.

After the future directions, a conclusion section should be added.

[Response]

We have added a conclusion section.

RR- accepted in general. however, i suggest developing it a little more, since it is too generic in the current version.

[Response2]

We have developed the conclusion section as suggested. (p.11, lines 484-485)

RRR- I THINK ANOTHER MAIN POINT IS THE REDUCTION OF SIDE EFFECTS, INCLUDING LONG-TERM COMORBIDITIES. THIS ASPECT SHOULD BE PART OF THE CONCLUSION AND, ACCORDINGLY, IN THE PREVIOUS SECTION, WHERE IT SEEMS THAT THIS IMPORTANT ASPECT/TOPIC HAS NOT MENTIONED AT ALL. THEN, I WOULD RECOMMEND THE AUTHORS TO ADD THIS ASPECT IN ONE OF THE PREVIOUS SECTION (MAYBE IN FUTURE DIRECTION) OR ADD A NEW ONE.

Comments on the Quality of English Language

SEE ABOVE
